# PixlMap: A generalisable pixel classifier for cellular phenotyping in multiplex immunofluorescence images

Rachel L. Baird[1], David Mason[2], Kai Rakovic[1,3,4], Fiona Ballantyne[1], Ian R. Powley[1], Anastasia Georgakopoulou[1,3], Lucy Hillary[3], Thomas G. Bird[1,3,5], Leah Officer-Jones[1]*, John Le Quesne[1,3,4]*

**1** Cancer Research UK Scotland Institute, Glasgow, United Kingdom, **2** Visiopharm A/S, Horsholm, Denmark, **3** School of Cancer Sciences, University of Glasgow, Glasgow, United Kingdom, **4** Department of Histopathology, Queen Elizabeth University Hospital, Glasgow, United Kingdom, **5** MRC Centre for Inflammation Research, The Queen's Medical Research Institute, University of Edinburgh, Edinburgh, United Kingdom

* l.officer-jones@crukscotlandinstitute.ac.uk (LOJ); John.LeQuesne@glasgow.ac.uk (JLQ)

## Abstract

Multiplexed methods for the detection of protein expression generate extremely data-rich images of intact tissue sections. These images are invaluable for the quantification and analysis of complex biology and biomarker development. However, their interpretation presents a considerable analytical challenge. Cell segmentation from images is a key bottleneck and a major focus of research activity in artificial intelligence. Most current methods depend initially on the use of a nuclear counterstain to identify nuclear boundaries, which is a relatively straightforward task. The cellular boundary is then assigned either by expansion of the nuclear outline, or by the use of membrane or cytoplasm-specific stains to delineate cell boundaries, or by some combination of the two. The task is critical, as inaccurate segmentation leads to information loss and data contamination from neighbouring cells. Increasingly sophisticated methods are being developed to address these issues, but each has its own shortcomings. We present an alternative method which is inspired by the fact that the assignation of a cellular phenotype 'by eye' does not depend upon the accurate identification of cell boundaries. We present an easy-to-use deep learning-based cellular phenotyping method which leverages this human capacity to assign phenotypes without segmenting the entire cell, and which can accurately phenotype cells based on nuclear segmentation alone. Using human ground truth annotations of entire cellular regions, we developed a classifier leveraging the U-Net architecture within a commercially available deep learning image analysis platform, but the principle is transferrable to any deep-learning framework. Crucially, training requires only a single example of each compartmental stain (nuclear/cytoplasmic/membranous). The resulting algorithm assigns class identities to cells with nuclear labelling alone, without the

**Data availability statement:** Code used to generate the figures in this publication is available from DOI https://doi.org/10.5281/zenodo.15593393. The Visiopharm APPs used to generate all figures are available from DOI https://doi.org/10.5281/zenodo.15593785. Murine images and layer data used in this publication are available at https://figshare.com/s/b9c355bd9b978b2ca564 Raw data, including images, relating to the LATTICe TMA cohort used in this publication are subject to NHSGGC Biorepository ethical approval. To access this data a biorepository application and MTA are required, please contact the Lead Researcher or the NHSGGC Biorepository manager (clare.orange@nhs.scot) for advice on the application process.

**Funding:** JLQ, LOJ, RLB, FB, LH, and IRP are funded by the Mazumdar Shaw Chair. KR is supported by a Pathological Society of Great Britain & Ireland and Jean Shanks Foundation clinical PhD fellowship (JSPS CPF 0422 04). TGB and AG were funded by an Accelerator Award from CRUK (Grant number: A26813) and CRUK Core funding (Grant numbers A17196, A31287 and CTRQQR-2021\100006). AG was funded by a CRUK HUNTER Studentship (Grant number: A28221). TGB, was funded by CRUK (Grant numbers: 23390 and DRCRPG-Nov22/100007). DM developed and maintains the Visiopharm software used in the study. DM was not involved in the data collection or data analysis parts of this study and declares no influence upon the results presented. The funders provided support in the form of salaries for all authors, but did not have any additional role in the study design, data collection and analysis, decision to publish, or preparation of the manuscript. The specific roles of these authors are articulated in the 'author contributions' section.

**Competing interests:** We have read the journal's policy and the authors of this manuscript have the following competing interests: DM is an employee of Visiopharm. All other authors declare no conflict of interest. There are no patents, products in development or marketed products associated with this research to declare. This does not alter our adherence to PLOS ONE policies on sharing data and materials.

need for whole cell expansion. The method is highly novel, broadly generalisable, and comparable in accuracy to intensity-based phenotyping methods, bridging the gap between inaccurate cellular segmentation and accurate phenotype generation.

## Introduction

Chromogenic immunohistochemistry (IHC) remains a gold standard clinical pathology test, revealing the expression of single proteins *in situ* to diagnose disease and stratify patients. However, newer methods using multiple colour channels to simultaneously quantify many target molecules are becoming increasingly common. This is largely driven by the need to understand the cellular composition and spatial interactions within complex tissue environments, such as the tumour microenvironment (TME) in cancer, as well as cellular niches in other disease types. This has driven the development and commercialisation of multiplex methods including multiplex immunofluorescence (mIF), various RNA detection methods, and hybrid multimodal approaches. These allow researchers to interrogate these disease types at the cellular level uncovering spatial interactions [1] and previously unknown phenotypes [2] in intact tissue samples. Such methods generate extremely data-rich images, but most analytical pipelines rely heavily on accurate cell segmentation to generate meaningful results. Accurate segmentation of multiplex images presents many challenges, and when done poorly errors can lead to misclassification and reduced accuracy, for example by combining information in overlapping or adjacent cells, therefore creating false phenotypes. Automated image analysis approaches are becoming essential, as manual methods, while potentially more accurate, become more laborious, subjective and time-consuming as image complexity grows. Automation removes subjectivity and the opportunity for human error, improving consistency whilst increasing throughput.

The current *de facto* standard approach used by almost all cell segmentation strategies relies upon the detection of a nuclear outline, typically achieved using a DNA stain such as DAPI, followed by an expansion step to include as much of the surrounding cytoplasmic area as possible. This permits the calculation of whole cell mean intensity per image channel. These values would then typically be used for cellular phenotyping, sorting cells into lineage categories by thresholding, clustering, or alternative computational techniques. Regardless of whether an intensity threshold or clustering technique is used to assign phenotypes [3] the foundational input remains the mean intensity per biomarker per cell. There have been various attempts to improve the accuracy of cell detection, including spillover correction [4], combinations of markers for cell segmentation [5,6] and the introduction of deep learning methods, which have led to the publication of numerous algorithms [7–9]. Cell segmentation using antibody/probe cocktails, although offering improved segmentation in several cell types, does not yet offer a universal solution. Currently the most effective methods combine several approaches, though optimal solutions typically need to be tailored to each experiment.

We have developed a novel deep learning approach, PixlMap, to assign accurate cellular phenotypes in a way which requires only nuclear segmentation without the need for whole cell segmentation at all. Whilst our bespoke analysis protocol package (APP) uses the deep learning tools in the Visiopharm digital image analysis platform, the same principle can be applied within any sufficiently powerful framework. Our method has been trained to recognise patterns of positivity for each of the major cellular compartments, i.e., cytoplasm, surface membrane, and nucleus, and assigns a resultant likelihood of belonging to a particular cell class to each individual pixel, based upon the broader local context. This permits accurate cellular phenotyping from assessment of just a small number of pixels within each cell, eliminating the need for accurate cell boundary determination and minimising even the requirement for accurate nucleus boundary segmentation.

## Materials and methods

### Multiplex immunofluorescence staining and image collection

**Human samples.** The LATTICe cohort was originally constructed under REC 14/EM/1159 (East Midlands REC), and the ongoing management of the resource by the Greater Glasgow and Clyde Biorepository was approved under an amendment granted by the Leicester South REC. Ongoing use of the collection is now managed under REC 16/WS/0207, biorepository approval number 644.

Multiplex images of lung adenocarcinoma (LUAD) tissue microarray (TMA) samples from the well-characterised LATTICe-A cohort [10] were used to train the deep learning classifier (Panels 1–4 in S1 Table). All multiplex assays were optimised on relevant control tissues using the Roche Ventana Discovery ULTRA (Roche Tissue Diagnostics). 4 µm thick sections were stained with multiplex immunofluorescent (mIF) panels as previously described [11–15]. Detailed information on the mIF assays and antibodies is provided in Supplementary S1 Table Individual core images in each mIF panel were collected at 20x magnification (except for panel 7 in S1 Table where core images were collected at 40x magnification) using the Akoya PhenoImager HT multispectral slide scanner (Akoya Biosciences) and spectrally unmixed using Inform software (Akoya Biosciences version 2.6.0).

For the deep learning training substrate, single fluorescent channels from panels 1–4 in S1 Table were extracted from mIF component images using QuPath (version 0.4.x) and individual channel images were subsequently imported into Visiopharm.

**Murine samples.** All animal experiments were conducted in accordance with UK Home Office licences (70/8891, PP0604995, 70/8646, 70/8468, and PP8854860) and in accordance with UK Animal (Scientific Procedures) Act 1986 and EU direction 2010. They were subject to review by the animal welfare and ethical review board of the University of Glasgow and the University of Newcastle upon Tyne. ARRIVE guidelines were followed for reporting of animal experiments, as previously described [16]. Single use of needles and non-adverse handling techniques were employed to minimize distress, pain and suffering. Mice were housed in a pathogen-free environment under standard conditions (19–22°C, 45–65% humidity, and 12 hours light-dark cycle) with access to food and water ad libitum. Environmental enrichment, including shredded paper, cardboard houses, wooden chews, and tunnels, was provided in all cages to encourage burrowing, natural hiding, and nesting behaviours. The following transgenic mouse strains were used: Ctnnb1$^{tm1Mmt}$ (*Ctnnb1*$^{ex3}$) [17], Gt(ROSA)26Sor$^{tm1(MYC)Djmy}$ (*R26*$^{LSL-MYC}$) [18]. Following ear-notch for identification purposes at weaning (3 weeks of age), ear tissue was sent to Transnetyx, Inc (Cordova, TN 38016) for genotyping using real-time PCR assay.

Male *Ctnnb1*$^{ex3/wt}$; *R26*$^{LSL-MYC/LSL-MYC}$ mice of mixed background were induced between 8–12 weeks of age, with AAV8. TBG.PI.Cre.rBG (AAV8-TBG-Cre) (Addgene, #107787-AAV8) as described previously [19]. Virus was diluted in sterile PBS to the desired working titre (6.4*10^8 GC/ml) and 100µl was administered to each mouse vial tail vein injection. Mice were sampled as day 120 time-point. All mice were humanely euthanised using $CO_2$ inhalation followed by cervical dislocation. Liver tissue was collected and placed in 10% neutral buffered formalin (NBF). After 24 hours, formalin was replaced with 70% ethanol prior to tissue processing and embedding in paraffin blocks.

4 µm thick murine sample sections were prepared and stained (Panel 5, S1 Table) as previously described [11–14]. Individual liver lobe regions were collected at 20x magnification using the Akoya PhenoImager HT multispectral slide scanner (Akoya Biosciences) and spectrally unmixed using Inform software (Akoya Biosciences version 2.6.0).

**Image analysis**

All image analysis was carried out in Visiopharm (version 2022.12.0.12865) using a physical Supermicro server running Windows 2016 Datacentre, with dual AMD EPYC 7713 processors, 256 GB RAM, Sabrent Rocket Q 8TB fast disk and four NVIDIA Quadro RTX 5000 GPU cards.

**Deep learning training.** PixlMap was trained on single-channel information extracted from mIF panels using the workflow shown in Fig 1. Training was performed on a virtualised server running Windows server 2016 Datacentre with a grid NVIDIA Tesla V100-PCIE-32Gb card Virtualised processors providing 32 cores and 64GB RAM. The magnification for training was set at 20x magnification at 0.5 µm per pixel. Ground truth regions were manually annotated using freehand image segmentation on single-channel images giving labelled examples of positivity for each cellular compartment, with negative regions labelled as background. The total training set included 26 training regions from 26 different LUAD patient cores, from 4 separate mIF panels. Markers used to train each cellular compartment were as follows: pan-cytokeratin (AE1/AE3), RPS6 (5G10), and p-eIF2a (D9G8) for cytoplasmic labelling; PD-L1 (E1L2N), CAIX (poly), CD4 (SP35), and CD8 (C8/144B) for membranous labelling; and FOXP3 (236A/E7) and Ki-67 (30−9) for nuclear labelling. The chosen markers were selected based on their robust and compartment-specific staining properties, ensuring high confidence in annotation. Additionally, LUAD was selected as a training substrate due to its well-characterised tumour microenvironment and availability of multiplexed data. All annotated training images were verified by a pathologist (KR, JLQ) prior to training the classifier.

Using our fully annotated training set, we developed a new Visiopharm APP using the deep learning module with the U-Net architecture. Due to all images in the training set containing just one feature, the architecture of the classifier was set to use 1 input channel. The size of the receptive field sent to the training network was set to 512 x 512 pixels (1 pixel = 0.5 µm$^2$). The training parameters included a learning rate of 1.0e-06 and a mini batch size of 2 to increase step-wise training accuracy without sacrificing speed. The APP underwent 539,000 iterations for training with the freeze depth being adjusted every 100,000 iterations for enhanced refinement. The final model achieved a loss function value of 0.035. The trained classifier outputs two features: Positive and Background and the resulting phenotypic heatmaps represent the probability that each pixel belongs to a specific cellular phenotype, with white areas indicating a high probability of belonging to a given phenotype. These probability values can be dynamically thresholded for downstream phenotyping analysis, enabling robust and adaptable cellular classification.

**Nuclear segmentation.** PixlMap was applied to previously classified images that were segmented into labelled nuclei with associated X/Y coordinates. Nucleus labelling was generated by re-training the pre-trained APP developed by Visiopharm (Nuclei Detection, AI (Fluorescence), version 2021.02.0.9284). To ensure accuracy, we manually annotated nuclei and nuclear boundaries on 1 mm TMA core images (n = 12 patient cores), providing additional ground truth data for training. The APP was further refined with 255,000 additional training iterations. This re-trained APP was subsequently applied to all images used in this study, ensuring consistent nucleus detection across the dataset.

**Assigning Phenotypes Using PixlMap Probability Heatmaps.** Nuclei were assigned to phenotypes based on probability heatmaps generated from PixlMap. We created a version of PixlMap for each phenotype that we wished to assign. Each of these APPs assigns nuclei phenotype labels (positive or negative) based on their heatmap intensity for marker of interest, using post-processing steps within Visiopharm. We defined an object as positive if 80–100% of the object's pixels had a positive probability of between 0.5 and 1.0, corresponding to a confidence level of 50–100%. The decision to set a threshold of 80% coverage or higher was made by observing the automated classifier against the ground truth. To accommodate mIF data, a separate APP was developed for each channel. While this modular approach creates

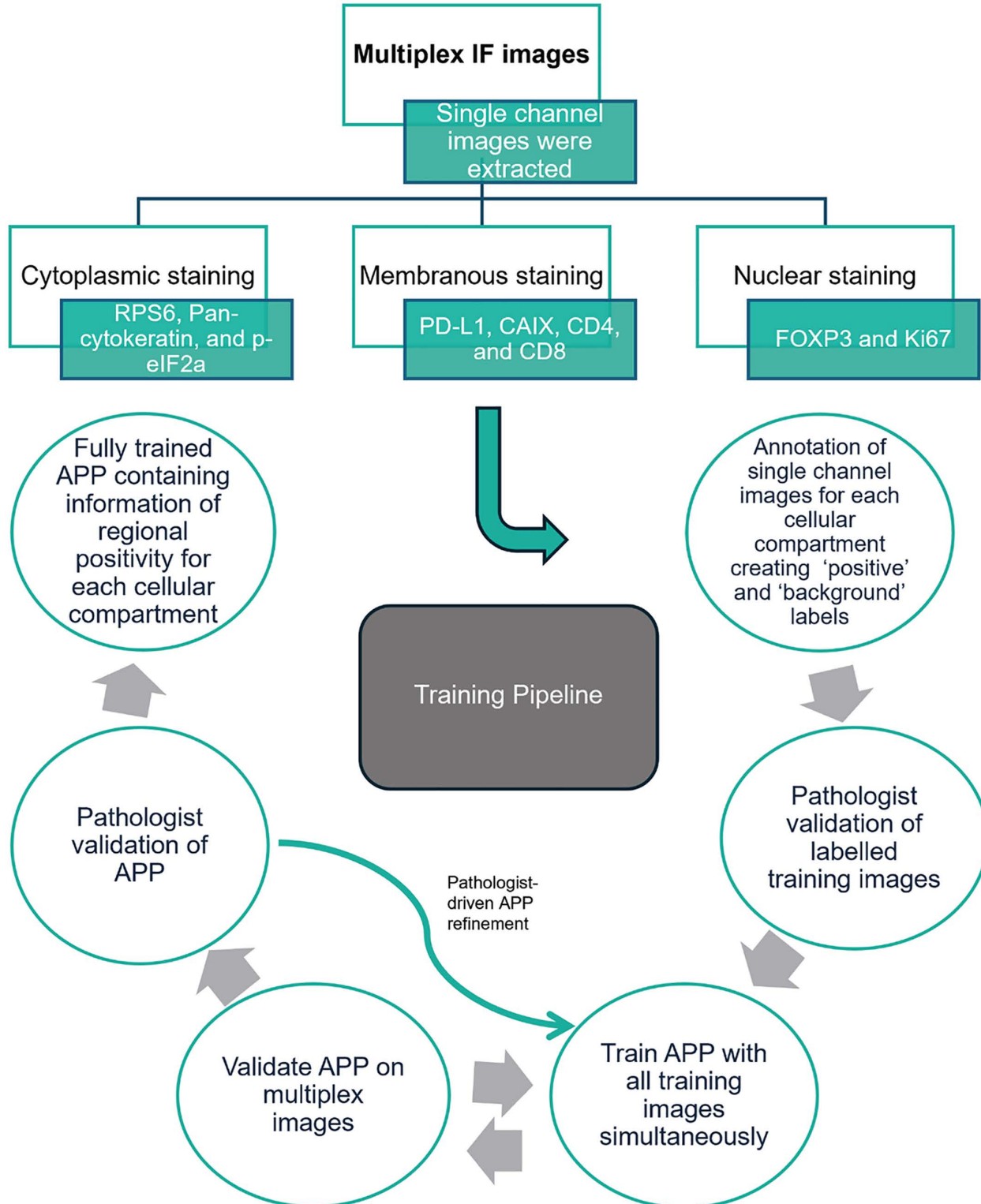

**Fig 1. Deep learning training pipeline.** A schematic map showing the deep learning training pipeline. The training set comprised regions from 26 different lung adenocarcinoma patient cores, from 4 separate experimental mIF panels. Markers used to train each cellular compartment were as follows: pan-cytokeratin, RPS6, and p-eIF2a for cytoplasmic labelling; PD-L1, CAIX, CD4, and CD8 for membranous labelling; and FOXP3, and Ki-67 for nuclear labelling. All training labels were verified by a pathologist prior to training.

separate APPs for each channel, its scalability allows for simultaneous analysis of multiple markers, for example, for a phenotyping project with five markers, this approach generates five distinct APPs, allowing independent evaluation of each marker.

**Determination of object sizes for accurate phenotyping.** In theory, cellular phenotypes could be assigned to heatmap objects of any size, from larger than an entire cell down to a single pixel in the centre of the nucleus. Crucially, our method assigns cell phenotype probabilities based on deep learning interpretation of 512 x 512 pixel (256 x 256 µm) regions to single pixels, so that a single pixel contains information about the surrounding region. This transference of information to the centre explains the ability of nuclear pixels in the centre of the cell image in the heatmap to provide information about cytoplasmic or membranous signals.

The smaller the object used to represent each cell, the less the computational cost, but the less heatmap information will be used. Optimal object size will ensure that phenotype assignment is maximally accurate and efficient.

Therefore, to determine the minimum object size for accurate phenotype assignment, we compared 4 different nuclear outline sizes to a ground truth dataset. A nuclear detection APP was applied within a region of interest of size 29,409 µm$^2$ to generate nuclear labels on a human LUAD image from mIF panel 2 (S1 Table). Each nuclear label was then outlined with an annotation to generate consistent nuclear X/Y coordinates for subsequent classification. The labelled image was assessed by a human pathologist to generate ground truth cytokeratin positivity, and 5 separate automated analyses were applied.

The five test groups were as follows:

1. Group 1: Whole nuclear outlines generated by the APP.

2. Group 2: Nuclear labels dilated by 3 pixels to include surrounding cytoplasm.

3. Group 3: Nuclear labels dilated by 8 pixels to capture additional cytoplasmic regions.

4. Group 4: Nuclear labels reduced to a single centre-of-mass pixel.

5. Group 5: Nuclear labels reduced to a single pixel centre-of-mass pixel and dilated by 3 pixels.

Nuclear dilation was performed during the post-processing stage of the Visiopharm APP design. Using the 'dilate' tool in post-processing steps, we specified the number of pixels by which each nucleus would be expanded in order to create the corresponding test group labels. Labels in all 5 groups were then changed to a new positive label based on the pan-cytokeratin probability heatmap from PixlMap when the probability exceeded 0.5 as described above. Results for all methods were compared to ground truth.

**Benchmarking.** To benchmark our method against the existing widely accepted phenotype methods, we compared PixlMap to the 'gold standard' manual assignment of phenotype identities by a pathologist, and to mean intensity-based phenotype generation which is a method built in to most commercially available platforms [20,21]. We applied a random region of interest (84,394 µm$^2$) to 16 LUAD core images of a 6-plex immune phenotyping assay (S1 Table Panel 2). We then used a nuclear detection APP as described above to generate nuclear labels (total n=9409) within all regions. All 16 images, including their nuclear overlays, were copied into their corresponding comparison folders (ground truth, mean intensity, and PixlMap) for phenotype assignment. For the ground truth dataset, we manually assigned phenotypes to cells according to their lineage-defining markers (pan-cytokeratin, CD8, CD4, CD68 or SMA) which was further reviewed by our expert multi-disciplinary tissue imaging team and a pathologist. Next, for each nuclear label, under section 'output variables' in the APP design, we generated mean intensity measures for each marker and a pathologist assigned a positive staining threshold by manual assessment of images to create the mean intensity dataset. All cells below the appropriate threshold were classed as negative. We also generated mean probabilities using PixlMap heatmaps at 10%, 25%, 50%, 75%, and 90% confidences, to create the PixlMap dataset. X/Y coordinates for each nucleus were generated

in all 3 datasets and used for performance metric analysis. Where PixlMap generated an incompatible phenotype, the cell was reclassified as 'unclassifiable'. Weighted performance metrics (F1 score, accuracy, precision, recall, AUC-ROC) and confusion matrices were calculated using scikit-learn v1.4.0 [22] and displayed with seaborn v0.13.2 [23] in python v3.10. The weighted F1 score, accuracy, precision, and recall provide a balanced evaluation of classification performance, accounting for potential class imbalances. AUC-ROC further quantifies PixlMap's discriminative power across phenotypes.

## Results

We trained PixlMap on regional staining information for each cellular compartment, using compartment-specific staining patterns to generate training labels (Fig 2). The training set included manually annotated examples of cytoplasmic, membranous, and nuclear staining patterns. For membranous and cytoplasmic markers, entire regions containing positive cells were labelled, including nuclei no matter their expression intensity; in this way, the channel positivity status of the relevant extranuclear compartment was applied to the nuclear area as well as the rest of the cell. For nuclear markers, only the nuclei themselves were labelled. By focusing on nuclear outlines, PixlMap simplifies phenotype assignment by removing the requirement for complex whole-cell segmentation, while still capturing staining information relevant to cellular classification. This methodology bypasses the computational complexity and potential inaccuracies associated with traditional whole-cell segmentation, enabling more robust and efficient cellular phenotyping.

Once PixlMap had shown stabilised deep learning convergence, we assessed classifier performance on test mIF images. PixlMap detected regions of positivity for all the markers within mIF images, providing accurate heatmaps which identify regions of positivity for cytoplasmic and membranous markers (Fig 3a and b, respectively) as well as positive

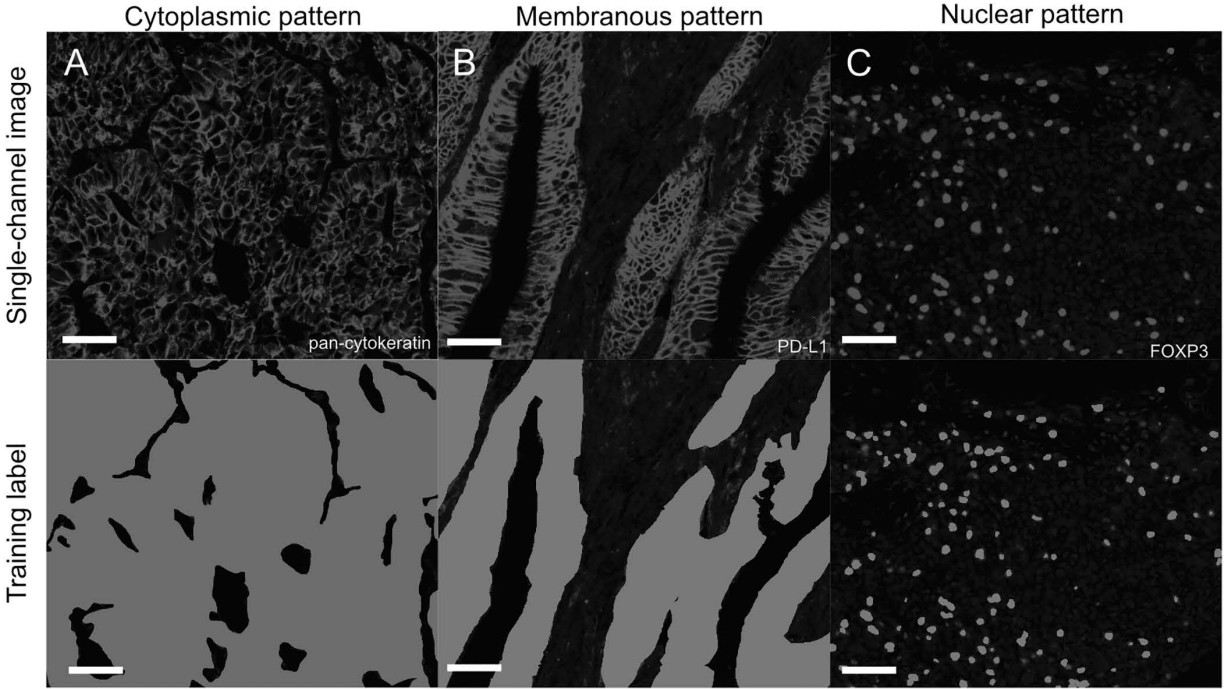

**Fig 2. Training PixlMap on human lung adenocarcinoma cores.** Examples of the training labels used to train PixlMap with each compartmental stain: cytoplasmic (pan-cytokeratin), membranous (PD-L1), and nuclear pattern (FOXP3), respectively from left to right (top), shown next to their corresponding manually annotated labels (bottom). Each training label highlights positive regions, and unlabelled regions were classed as background. Training labels were generated from single channel images generated from mIF panels 1-4 (S1 Table). Scale bar = 50 μm.

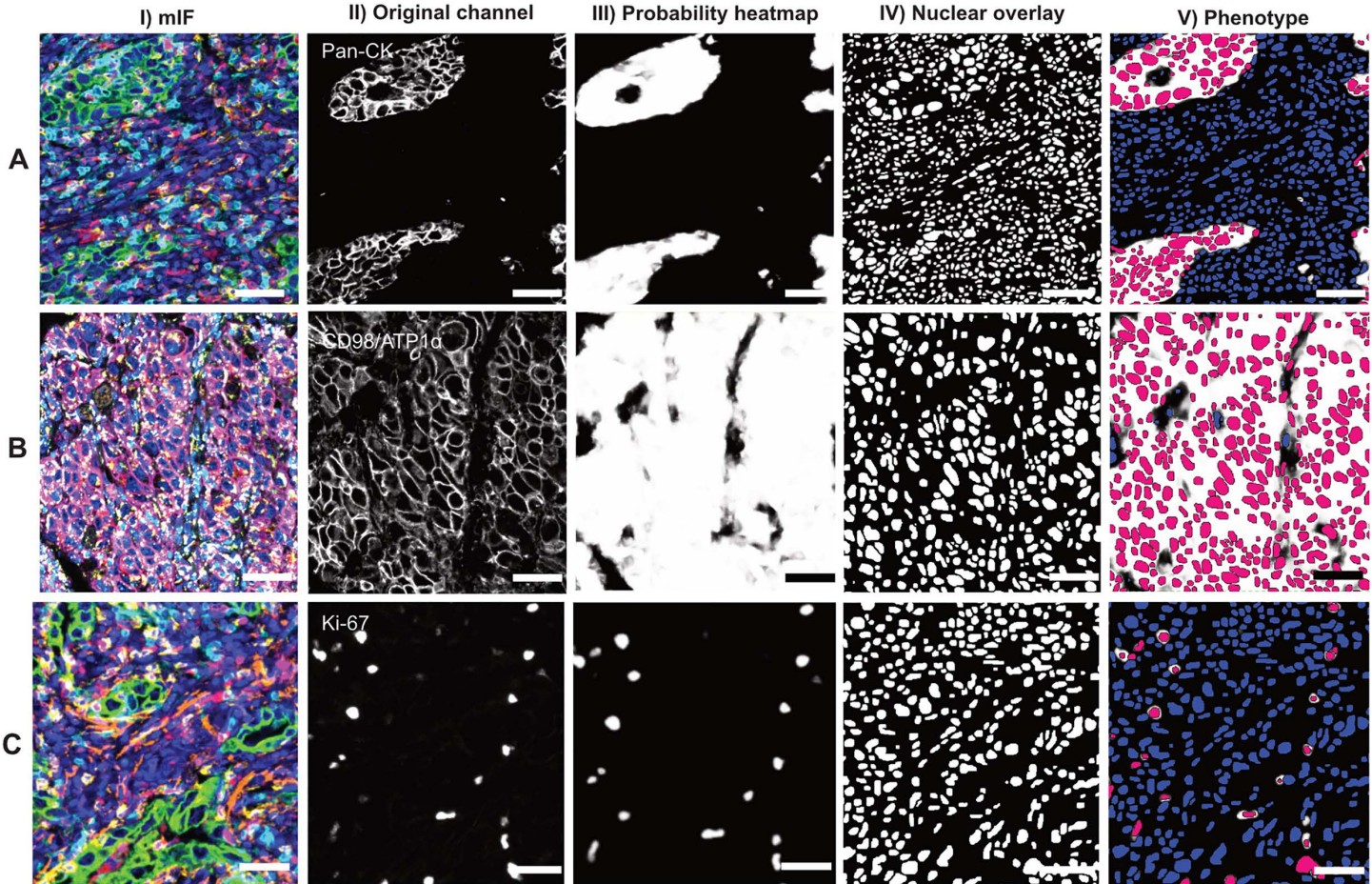

**Fig 3. PixlMap extracts probability heatmaps from multiplex images to classify cells. A. PixlMap identifies nuclei in cells with cytoplasmic staining.** I.) A mIF image (panel 2 S1 Table) of human LUAD containing markers CD4 (cyan), CD8 (yellow), Ki-67 (magenta), CD68 (red), pan-cytokeratin (green), and smooth muscle actin (Orange), with a DAPI counterstain (blue). II.) Pan-cytokeratin channel of interest. III.) Pan-cytokeratin pixel probability heatmap (white, positive = 1, black, negative = 0). IV.) Nuclear labels. V.) Positive (magenta), and negative (blue) nuclear labels using PixlMap pan-cytokeratin heatmap, with a 50% confidence. Scale bar = 50 μm. **B. PixlMap identifies nuclei in cells with positive membranous staining.** I.) A mIF image (panel 7 in S1 Table) of human LUAD containing markers UBF1 (light blue), Calnexin (light pink), CD98/ATP1α cocktail (magenta), GLB1 (cyan), RPS6 (orange), GOLG1 (red), ATP5A (yellow), pan-cytokeratin (not shown), and a DAPI counterstain (blue). II.) CD98/ATP1α channel of interest. III.) CD98/ATP1α cocktail pixel probability heatmap (white, positive = 1, black, negative = 0). IV.) Nuclear labels. V.) Positive (magenta), and negative (blue) nuclear labels using PixlMap CD98/ATP1α heatmap, with a 50% confidence. Scale bar = 50 μm. **C. PixlMap identifies positive nuclei.** I.) A mIF image (panel 2 in S1 Table) of human LUAD containing markers CD4 (cyan), CD8 (yellow), Ki-67 (magenta), CD68 (red), pan-cytokeratin (green), and smooth muscle actin (Orange), with a DAPI counterstain (blue). II.) Ki-67 channel of interest. III.) Ki-67 pixel probability heatmap (white, positive = 1, black, negative = 0). IV.) Nuclear labels. V.) Positive (magenta), and negative (blue) nuclear labels using PixlMap Ki-67 heatmap, with a 50% confidence. Scale bar = 50 μm.

nuclei (Fig 3c). Phenotypes are assigned by first applying a nuclear detection APP to generate nuclear labels before using the positivity heatmap from PixlMap to assign phenotypes to each label with specified confidence limits.

We set out to gauge the optimum label requirement size to correctly classify pan-cytokeratin positive cells by comparing 5 test groups to manually classify pan-cytokeratin positive cells (Fig 4). Test groups were as follows (total cells n = 401 per test group): Group 1 included nuclear-sized labels only; group 2 included nuclear labels + 3-pixel expansion; group 3 included nuclear labels +8-pixel expansion; group 4 included single-pixel in the centre of the nucleus, and group 5 included a 3-pixel-sized nucleus. We found that whole nuclear labelling was optimal for phenotype assignment (Fig 4E)

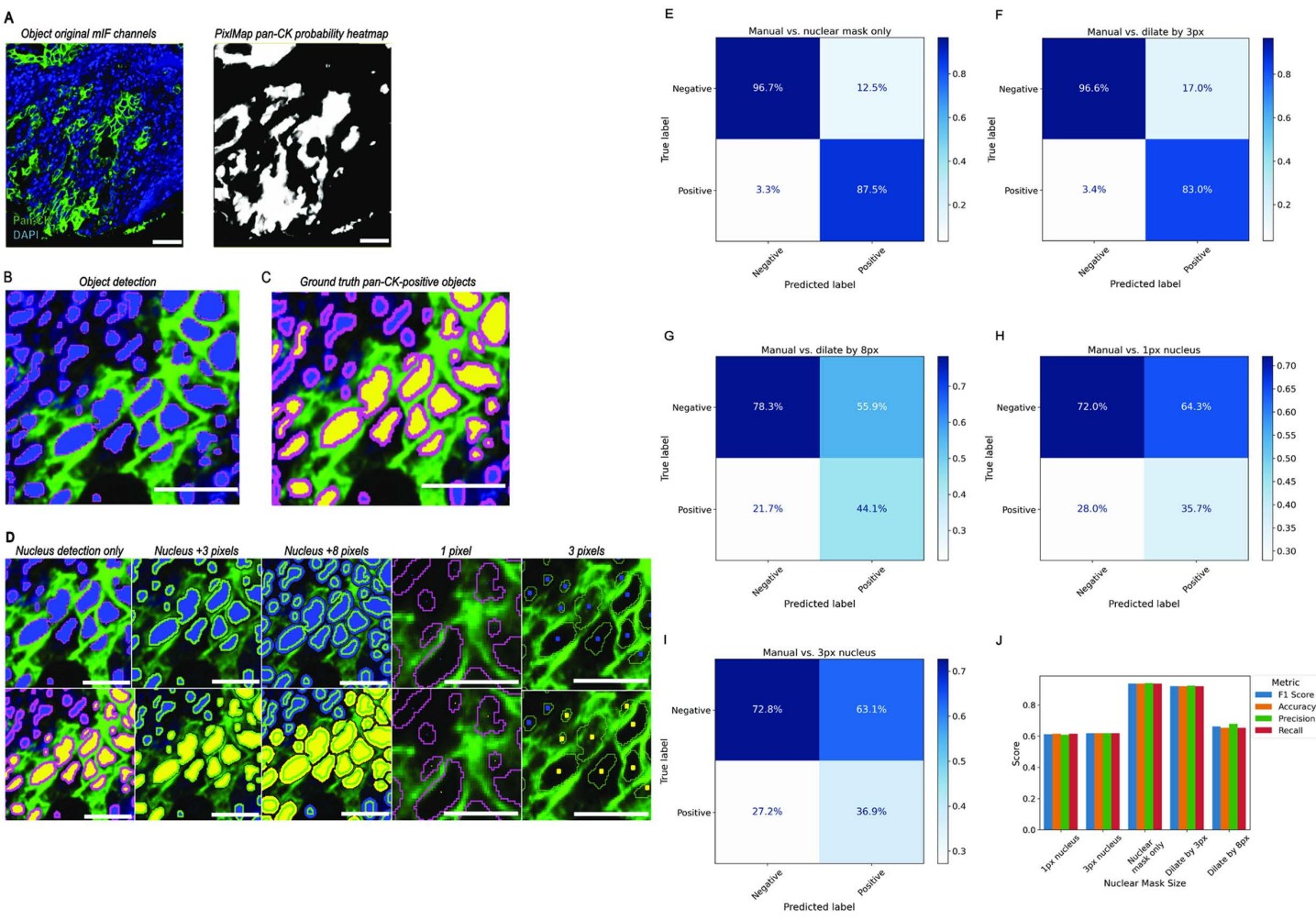

**Fig 4. PixlMap can correctly classify cells without the need for whole cell expansion. A.** A mIF image of panel 2 in S1 Table showing only pan-cytokeratin (green) and DAPI (blue) (left), and PixlMap's pan-cytokeratin probability heatmap from the mIF image (right). Scale bar = 50 μm. **B.** A nuclear detection APP was applied in a default region of interest (29,409 μm²) Nuclei (blue) were outlined with an annotation (magenta) for X/Y calculations to ensure each X/Y coordinate stayed the same in all test groups. The image was then copied with its layer data into different folders for each test group (n = 5). Scale bar = 50 μm. C. Ground truth group: blue labels were manually assigned a pan-cytokeratin-positive label (yellow) by a pathologist (n = 121 positive, n = 280 negative). Scale bar = 50 μm. **D.** Comparison groups: blue labels in each comparison group (top) were changed to a pan-cytokeratin-positive label (yellow) (bottom) using the PixlMap pan-cytokeratin heatmap with 50% confidence (0.5): nuclear-size (left), nuclear-size + 3 pixels (middle left), nuclear-size + 8 pixels (middle), single pixel-sized (middle right), 3-pixel nucleus (far right). Scale bar = 50 μm. **E-I.** Confusion matrices comparing ground truth pan-cytokeratin assignment to PixlMap predictions by nuclear label size. Nuclear size only, nuclear label +3 pixels, nuclear label +8 pixels, 1-pixel label, and a 3-pixel label, respectively. Percentages normalised to predications. **J.** Bar plot showing weighted performance statistics for each label size (weighted F1 score, accuracy, precision, recall).

and was comparable to the ground truth group with the most precision and accuracy, compared with the other groups (Fig 4F-J). Therefore, from this point on, we used whole nucleus labels for analysis.

We next tested the ability of PixlMap to correctly assign phenotypes based on multiple markers expressed in the same compartment of different cell types (Fig 5a). PixlMap can extract probability heatmaps for CD4 and CD8 and use them to assign phenotypes, highlighting that PixlMap can distinguish between these T-cell types despite their identical morphologies. Furthermore, PixlMap can correctly classify cells with slightly different morphologies from those that were included in

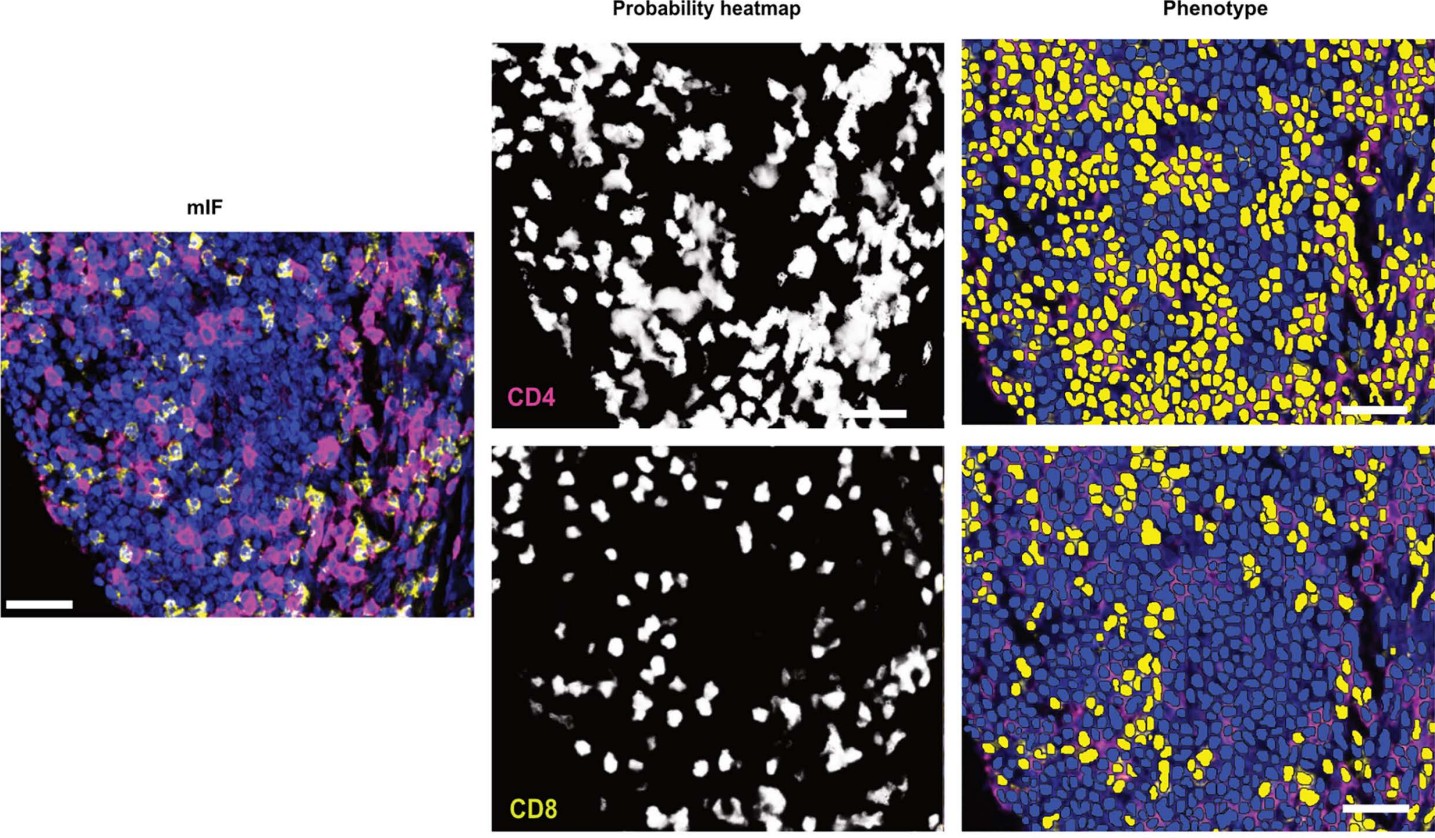

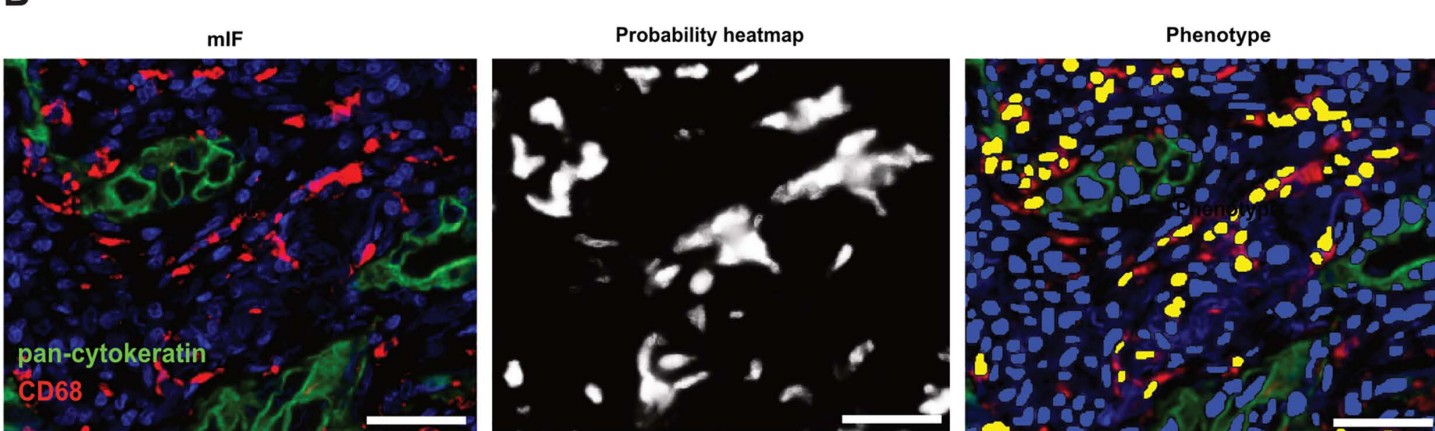

**Fig 5. PixlMap can discriminate between phenotypes of various morphologies. A.** mIF panel 2 in S1 Table in human LUAD with CD4 (magenta), CD8 (yellow), and DAPI (blue) channels turned on only for visual (left). Middle shows the CD4 and CD8 pixel probability heatmaps extracted from the multiplex image by PixlMap (white, positive = 1, black, negative = 0). Right shows the nuclear overlays (blue) where nuclei were changed to a positive (yellow) label based on the corresponding CD4 (top) and CD8 (bottom) PixlMap probability heatmaps with 50% confidence (0.5). Scale bar = 50 μm. **B.** Highlights mIF panel 2 in S1 Table but with the pan-cytokeratin (green), CD68 (red), and DAPI (blue) channels turned on only for visual (left). Middle depicts the CD68 pixel probability heatmap extracted from the mIF image by PixlMap (white positive = 1, black, negative = 0). Left highlights the nuclear overlays (blue) were changed to a positive (yellow) label based on the corresponding CD68 probability heatmap with 50% confidence (0.5). Scale bar = 50 μm.

the training set. For example, CD68 membrane positivity in macrophages has a quite distinct pattern from all training cell types. Despite this, PixlMap was able to accurately phenotype CD68 positive cells (Fig 5b).

To further validate PixlMap against ground truth, we focussed on lineages defined by single markers. We compared manual annotations from a pathologist to PixlMap predictions of the following cell types: CD4-positive T-cells, CD8-positive T-cells, CD68-positive macrophages, pan-cytokeratin-positive epithelial cells, and SMA-positive myofibroblasts in human LUAD (S1 Fig 1) using panel 2 (S1 Table). We noted that PixlMap failed to accurately identify SMA positive cells, so we focussed on measuring performance on non-spindled cells for the remainder of this task, i.e., using the same markers in S1 Fig 1, but excluding SMA in the analysis (Fig 6). Our results further justify using PixlMap to assign class identities to labels of nuclear size, as it was highly comparable to manual phenotyping of these cells with multiple confidence limits (Fig 6A-F). Next, we compared PixlMap and mean intensity-based phenotyping and compared the respective outputs to the same ground truth labels (Fig 6H). Our results highlight that we can phenotype cells more accurately using PixlMap with nucleus segmentation only, than with the standard whole cell expansion-mean intensity method for phenotyping.

To evaluate whether our APP would work across different species and tissue types, we next tested PixlMap's human tissue-trained phenotyping capabilities on murine mIF images (Fig 7). We ran a nuclear detection APP on images from an immunofluorescent panel (Fig 7A-D) (S1 Table, panel 5) of murine liver samples (n = 6), comprising of a total of 6208 cells. Nuclear labels (blue) were reclassified to a positive label (magenta) using PixlMap's heatmap when the probability exceeded 0.5 (i.e., 50%) for the respective phenotypic input. Given PixlMap's success in assigning single lineage pheno-types to cells in the human LUAD panel, we extended its application to murine liver samples to evaluate its performance in assigning single lineage phenotypes in this new context. Interestingly, despite being trained on human tumour tissue, we found that PixlMap can extract accurate positivity heatmaps for single lineage markers (CD4, CD8, and GS), and such heatmaps can be used to correctly assign class identities with high confidence compared to human ground truth assign-ment (Fig 7E-J).

We revisited the spindled cell population to further explore PixlMap's utility on cells with this morphology. We predict that PixlMap's lack of ability to phenotype SMA positive cells is due to spindle-shaped cells being absent in the training data and therefore further explored this concept. We have shown that PixlMap has a limited capability to correctly extract probability heatmaps from fibroblast markers in a fibroblast-specific mIF panel (S1 Table Panel 6); HLA-DR, FAP, Podo-planin, and IL-6 (Fig 8C I-IV), and will need further training for SMA heatmap extraction (Fig 8C V). Therefore, additional training to include different fibroblast morphologies will be required for accurate fibroblast phenotype assignment from PixlMap.

In summary we have developed a method which can easily assign single lineage phenotype cell classes to circular cells. PixlMap offers a more efficient, accurate, and scalable method for the phenotyping of these cells, reducing the complexity of traditional segmentation approaches while maintaining robust performance.

## Discussion

Accurate classification of cells in mIF images is essential to unlock the full wealth of biological information that they contain. Current phenotyping methods rely heavily on accurate cellular segmentation, and while methods to improve this continue to evolve, there is no universally trusted segmentation method. There are numerous factors affecting the quality of tissue, and, therefore, cellular segmentation, including cold ischaemic times, fixation times, and sample age. In addi-tion, the unique biological signatures of each patient's disease introduce variance between tumour tissues, making a stan-dardised approach to cell classification challenging. Consequently, accurately segmenting these cells to gain knowledge of highly diverse and complex micro-environments remains a significant bottleneck. Cell expansion methods from nuclear masks are commonly used for mean intensity-based phenotyping methods. Without knowledge of the true cell boundaries, a method of this sort is necessary to ensure that extranuclear information is not wasted. While such approaches generally improve the ability to phenotype cells, they are vulnerable to the heterogeneity of cell shapes and sizes, which cannot be

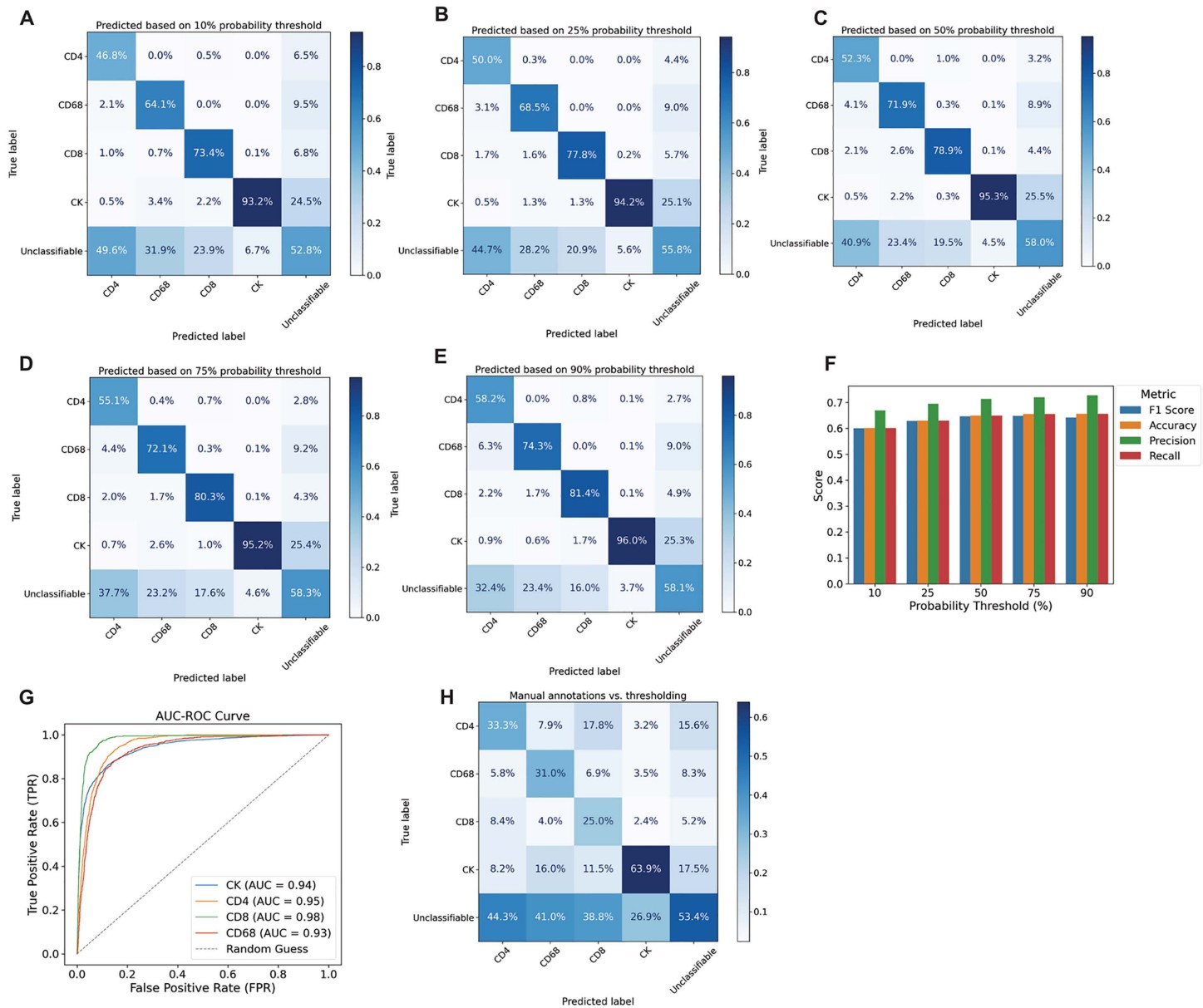

**Fig 6. PixlMap phenotyping is comparable to pathologist ground truth. A-E.** Confusion matrices comparing ground truth assignment of pan-cytokeratin (CK), CD4, CD8, and CD68 to PixlMap predictions by confidence thresholds. Percentages normalised to predications. **F.** Bar plot showing weighted performance statistics for single lineage phenotyping for CK, CD4, CD8, and CD68 (weighted F1 score, accuracy, precision, recall). **G.** AUC-ROC comparing PixlMap to ground truth single predications for CK, CD4, CD8, and CD68. **H.** Confusion matrix comparing ground truth manual annotations provided by a pathologist and threshold-based phenotyping using mean pixel intensities of each marker. True = ground truth and predicted is thresholding. Percentages normalised to predications.

accommodated by a simple dilation of the nuclear envelope. Newer methods which include adding extra markers for cellular boundaries into mIF assays, singly or as cocktails, can aid cellular segmentation. However, for lower-plex analysis, this reduces the number of experimental proteins that can be used at once, there is no universal cell boundary cocktail for all cell types [5,6,24] and some element of 'blind' nuclear expansion is always necessary.

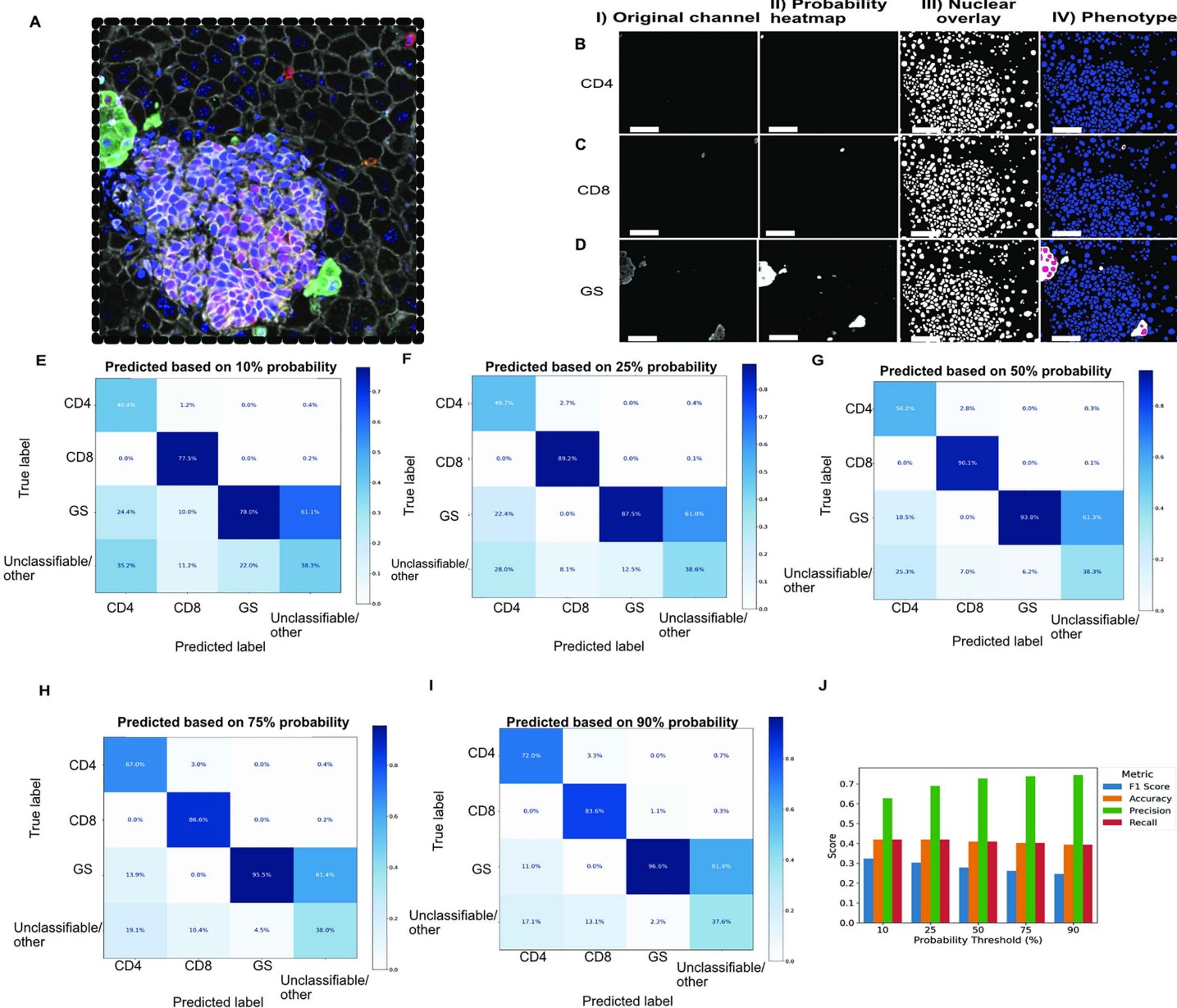

**Fig 7. PixlMap can extract probability heatmaps from murine liver samples. A.** PixlMap can extract probability heatmaps and classify cells in murine mIF images despite only being trained with human tissue images. The mIF panel (panel 5 in S1 Table) shown consists of GS (green), FOXP3 (not present), CD8 (red), Granzyme B (orange), CD45 (lilac), CD4 (cyan), c-Myc (magenta), β-Catenin (white), and a DAPI counterstain (blue). Scale bar = 50 μm. **B.** CD4 I.) Channel of interest II.) PixlMap heatmap III.) Nuclear labels IV.) CD4 positive (magenta), and CD4 negative (blue) based on a 50% PixlMap confidence. Scale bar = 50 μm. **C**. CD8 I.) Channel of interest II.) PixlMap heatmap III.) Nuclear labels IV.) CD8 positive (magenta), and CD8 negative (blue) based on a 50% PixlMap confidence. Scale bar = 50 μm. **D.** GS I.) Channel of interest II.) PixlMap heatmap III.) Nuclear labels IV.) GS positive (magenta), and GS negative (blue) based on a 50% PixlMap confidence. Scale bar = 50 μm. **E-I.** Confusion matrices comparing ground truth assignment of CD4, CD8, and GS to PixlMap predictions by confidence thresholds. Percentages normalised to predications. **J.** Bar plot showing weighted performance statistics for single lineage markers CD4, CD8, and GS (weighted F1 score, accuracy, precision, recall).

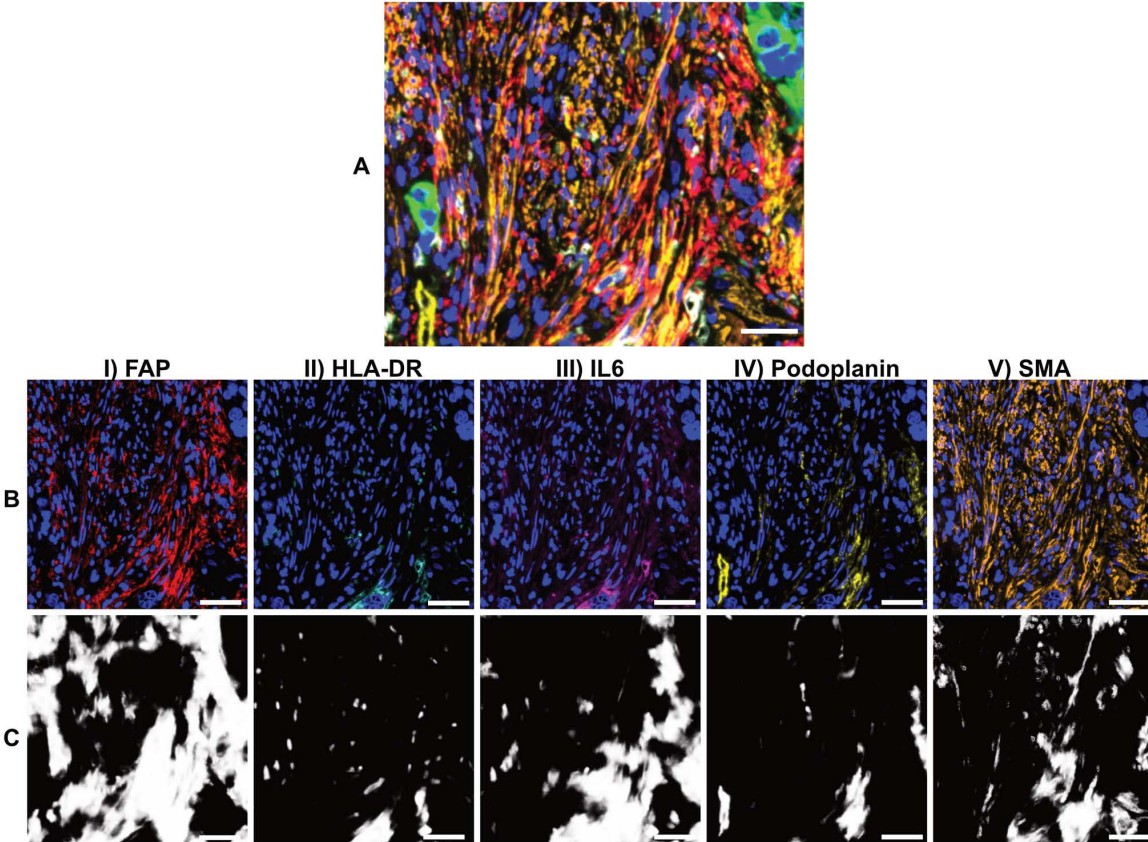

**Fig 8. PixlMap requires additional fibroblast morphology training for accurate fibroblast phenotyping. A.** mIF panel 6 in S1 Table with original mIF image of human LUAD with FAP (red), HLA-DR (cyan), IL6 (magenta), Podoplanin (yellow), SMA (orange), pan-cytokeratin (green), and DAPI (blue). Scale bar = 50 μm. **B.** I-V.) Original mIF channels showing FAP (red), HLA-DR (cyan), IL6 (magenta), Podoplanin (yellow), and SMA (orange), respectively, with DAPI (blue). Scale bar = 50 μm. **C.** PixlMap probability feature heatmap for the corresponding channels above in **B** as an input (white positive = 1, black, negative = 0). Scale bar = 50 μm.

Unlike other, more complex, deep learning methods which depend upon cellular segmentation [9], PixlMap requires limited hours spent annotating small datasets, making use of a highly accessible and well-characterised U-Net deep learning tool available in proprietary software [25]. Our simple training approach, using single channel information from mIF images to generate probability heatmaps generates an APP that is highly generalisable and can integrate information from multiple channels with multiple subcellular expression patterns. Crucially, we have shown that our method obviates the need for expansion of nuclear outlines to approximate cellular outlines, which is a significant source of error in classical approaches, and therefore does not require additional cell membrane markers for accurate phenotype assignment. This is achievable because our training data contains staining patterns of markers expressed in each major cellular compartment. By including nuclear areas in non-nuclear training annotations, the resulting heatmaps accurately predict cell phenotypes from nuclear labels even if the nuclei themselves are not stained. Our validation confirms that nuclear areas are sufficient to assign phenotypes to whole cells, and highly comparable to human assignment of phenotypes. Furthermore, we have shown that in our mIF datasets, increasing the object size for phenotype assignment (i.e., dilating from a nuclear label to generate an approximate whole cell label), is counter-productive, and even decreases the accuracy of phenotyping.

Cellular phenotyping based on mean pixel intensities of each marker is the current standard for phenotype assignment. However, our method is more nuanced, with training which includes real-world staining patterns. Such staining patterns can often be quite weak in absolute terms while still being convincingly positive to the human interpreter based on the pattern or staining quality; indeed, this subtle distinction is a key part of clinical immuno-histochemical interpretation. This approach, therefore, should partly overcome the need for extensive normalisation methods in downstream analysis for large datasets with high staining variability. Our benchmarking results have shown superiority of PixlMap phenotype assignment over the standard method of generating mean intensities for cellular compartments. However, like all microscopy-based imaging methods, our method still requires, to an extent, good tissue quality to generate meaningful data to accurately label nuclei or objects of interest. Any method, however, that accurately labels the centre of a cell or nucleus can be deployed for PixlMap to accurately assign cell class identities.

PixlMap shows considerable robustness. Its value is not limited to human tissue images, despite only using human tissue annotations for training on LUAD. We have demonstrated that PixlMap correctly assigns class identities across species and unseen tissue types. We evaluated this in murine liver tissue, further proving that our method is highly generalisable. While we observed reduced performance in murine tissue compared to LUAD, we demonstrate the transferability of much of the performance across species, with considerable promise for future refinement of the approach with species-specific training data.

Regions of diverse lymphoid structures are densely packed with cells and typically represent problematic areas both for cell and nuclear segmentation. Therefore, it is imperative that training data are accurate, especially with high heterogeneity in cell populations with varying object sizes. We found that our algorithm can overcome this issue and phenotype T cells in lymphoid aggregate structures with high accuracy using only nuclear labelling, when compared to manual phenotyping.

Despite Visiopharm being used for all our image analysis in this study, our method is replicable in other software including open systems. Given that U-Net is a well understood and deployed network [26], our method should be replicable with U-Net in Python/ FOSS software with the same level of accuracy.

mIF images contain an abundance of useful information which have obvious potential in facilitating diagnostics and personalised medicine. New quantitative spatial assays are likely to be crucial as biomarkers for the correct positioning of new therapies, including anti-cancer immunotherapies and antibody-drug conjugates (ADCs) [27]. However, before these biomarkers can be implemented in the clinical setting, there are many obstacles to overcome such as assay standardisation, the choice of an appropriate implementation model, quality assurance, cost, data storage, and data complexity [28]. Clinical decision making may also require pathologists to undergo additional training to interpret such complex images [29]. Automated digital pathology workflows have potential to significantly streamline the process. We show that the PixlMap APP approach may overcome some major challenges in these digital pathology workflows, being highly robust in cellular phenotyping in the crowded tumour microenvironment, with the potential to improve patient stratification in a broadly applicable manner.

Despite the robustness of our APP in different tissue types and species, it still has some limitations to overcome. Due to extremely non-circular cells not being included in the training set, PixlMap will probably require an additional morphology category to assign phenotypes to fibroblasts and other elongated cells. Phenotyping fibroblasts *in situ* already presents significant biological and technical challenges. One major issue is the absence of a definitive, universal marker that uniquely identifies fibroblasts and distinguishes them from other cell types [30]. Therefore, relying on a single marker for mean intensity-based phenotyping may introduce inaccurate or misleading information. Additionally, fibroblasts exhibit substantial variability in their morphology—while they are often described as long and thin, not all fibroblasts have this feature, making morphological identification also unreliable in many cases [31]. Despite the lack of prior knowledge of fibroblast staining or morphology, PixlMap did perform better than expected at this challenging task. As a result, we believe, by

training PixlMap on single channels containing fibroblast staining pattern information, we may be able to refine the detection of non-circular cells, as we need only a nuclear marker to assign class identities with PixlMap.

Similarly, macrophages do not have a consistent cellular shape, with their boundaries being unclear and discontinuous [32]. We have shown that PixlMap was able to phenotype macrophages in our dataset, however not as accurately as other cell types. We believe this is probably due to endogenous low-level CD4 signal in these cells that may have confused the algorithm. This could be ameliorated by the provision of additional macrophage-rich training images.

Furthermore, PixlMap will require further training on images generated by alternative staining and imaging platforms (Akoya PhenoCycler, Lunaphore COMET) due to the difference in autofluorescence removal in these imaging modalities [33–35]. Each of these platforms also processes images differently, however, a cross-platform training approach could overcome this in future versions.

## Conclusions and future work

PixlMap represents a paradigm shift in phenotyping approaches by eliminating the need for whole-cell segmentation, paving the way for simpler, more adaptable methods in tissue analysis. This work highlights that whole cell estimation by nuclear expansion is no longer necessary for accurate phenotype assignment using our method. With PixlMap, cellular phenotypes can be assigned using only nuclear segmentation. This approach simplifies the process of cellular phenotyping while maintaining high accuracy, leveraging our well-trained U-Net APP that assigns whole cell phenotypes directly to nuclei without the need to expand the entire cell structure. Moreover, the inclusion of confidence limits helps tailor the APPs predictions to different situations, depending on how much uncertainty is acceptable in a given context.

Future directions include further refining our APP to include examples of cells with different morphologies, e.g., fibroblasts. While others use known fibroblast markers to segment fibroblasts [36] and phenotype them based on their expression levels of specific fibroblast markers [37], there is scope for using our method for fibroblast phenotyping. We intend to further develop PixlMap as a cross-platform classifier by incorporating images from all of our mIF imaging platforms into our training data. Furthermore, we plan on testing the application of PixlMap to RNA/spot-based detection methods by including regions of cells with punctate signal in our training set until the desired stabilised convergence.

## Supporting information

**S1 Fig. PixlMap requires additional morphology training for fibroblast phenotyping. A-E.** Confusion matrices comparing ground truth assignment of pan-cytokeratin, CD4, CD8, CD68, and SMA to PixlMap predictions by confidence thresholds. Percentages normalised to predications. **F.** Bar plot showing weighted performance statistics (F1 score, accuracy, precision, recall).
(TIFF)

**S1 Table. mIF staining protocol parameters for the Ventana Discovery Ultra.**
(DOCX)

**S1 Data. Script for single-channel image extraction from mIF images.**
(TXT)

## Acknowledgments

This paper was critically reviewed by Catherine Winchester (CRUK Scotland Institute).

We would like to thank Peter McHardy (CRUK Scotland Institute) for his support with Visiopharm infrastructure throughout this project. We would also like to thank Kara Luckett, Silvia Martinelli, and Cat Ficken from the Le Quesne lab at the CRUK Scotland Institute for their support with image acquisition and data curation throughout this project.

We acknowledge the support of NHS Research Scotland (NRS) Greater Glasgow and Clyde Biorepository for their ongoing management of the LATTICe TMA cohort.

## Author contributions

**Conceptualization:** David Mason, Leah Officer-Jones, John Le Quesne.

**Formal analysis:** Kai Rakovic.

**Funding acquisition:** Thomas G Bird, John Le Quesne.

**Investigation:** Rachel L Baird, Kai Rakovic, Anastasia Georgakopoulou.

**Methodology:** Rachel L Baird, David Mason, Ian R Powley, Leah Officer-Jones.

**Resources:** Anastasia Georgakopoulou, Thomas G Bird, John Le Quesne.

**Supervision:** Leah Officer-Jones, John Le Quesne.

**Validation:** Rachel L Baird, Kai Rakovic.

**Visualization:** Rachel L Baird, Kai Rakovic, Fiona Ballantyne, Ian R Powley, Lucy Hillary.

**Writing – original draft:** Rachel L Baird, Leah Officer-Jones.

**Writing – review & editing:** David Mason, Kai Rakovic, Fiona Ballantyne, Ian R Powley, Anastasia Georgakopoulou, Thomas G Bird, John Le Quesne, Lucy Hillary.

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
