## [Decision Letter · Decision Letter 0]

28 Apr 2025

Dear Dr. Le Quesne,

Thank you for submitting your manuscript to PLOS ONE. After careful consideration, we feel that it has merit but does not fully meet PLOS ONE’s publication criteria as it currently stands. Therefore, we invite you to submit a revised version of the manuscript that addresses the points raised during the review process.

We look forward to receiving your revised manuscript.

Kind regards,

Jordan Robin Yaron, Ph.D.

Academic Editor

PLOS ONE

Journal Requirements:

“I have read the journal's policy and the authors of this manuscript have the following competing interests:DM is an employee of Visiopharm. All other authors declare no conflict of interest.”

We note that one or more of the authors are employed by a commercial company: Visiopharm

6. We note that you have indicated that there are restrictions to data sharing for this study. PLOS only allows data to be available upon request if there are legal or ethical restrictions on sharing data publicly. For more information on unacceptable data access restrictions, please see http://journals.plos.org/plosone/s/data-availability#loc-unacceptable-data-access-restrictions.

The number of images as well as cells/image that were analyzed for each experiment.Details on how mean intensity data were generated, thresholding approaches, expansion methods, and a representative mask from the images for each marker.Direct comparison of actual cell number results from manual evaluation, PixlMap, and traditional methods for 2-3 markers, along with number of images and approximate number of cells.Address, explain, or fix the issue with field/zoom representation in Figure 4A.Address in greater detail potential issues with fibroblast detection.Similarly, address potential accuracy issues in identifying CD68+ cells in Figure 5B.Address whether surface markers are more accurately identified versus cytoplasmic markers.Address weakness of metrics in liver tissue comparisons.Clarify language throughout regarding manual versus threshold-based phenotyping.

Reviewers' comments:

Reviewer's Responses to Questions

**Comments to the Author**

1. Is the manuscript technically sound, and do the data support the conclusions?

Reviewer #1: Partly

Reviewer #2: Partly

Reviewer #3: Yes

2. Has the statistical analysis been performed appropriately and rigorously?

Reviewer #1: Yes

Reviewer #2: Yes

Reviewer #3: N/A

3. Have the authors made all data underlying the findings in their manuscript fully available?

Reviewer #1: Yes

Reviewer #2: No

Reviewer #3: Yes

4. Is the manuscript presented in an intelligible fashion and written in standard English?

Reviewer #1: Yes

Reviewer #2: Yes

Reviewer #3: Yes

Reviewer #1: While the paper is well presented, it significantly lacks technological novelty. Given the recent advances in generative models, and other advanced mathematical tricks (contrastive learning, different loss functions such as focal loss, ...), I do not entirely subscribe to the idea that you would need significant annotations for this task. In fact, a fairly simple CNN designed with the correct parameters (optimal receptive field) and loss functions could very well be suited here. The article does not talk about the technically novel aspects. I would expect large vision models (such as SAM) to atleast provide a initial ground truth for learning tasks. Unfortunately, I do not think this article is technically solid enough for being published. The methods used (basically a U-Net) are dated, and there's little innovation in terms of architectural design or modelling.

Reviewer #2: The number of images (and cells/image) analyzed should be provided for each experiment.

Since the performance of the approach is being directly compared to existing approaches (mean intensity), more details are needed on how the mean intensity data were generated (Figure 6H). What thresholding approach was employed? What expansion method was used? Show a representative mask from the images for each marker.

Provide actual results in terms of cell numbers of different lineages/markers as determined by manual evaluation, PixlMap, traditional analysis methods. Do this for at leash 2 or 3 markers and indicate the number of images and approximate number of cells.

In Figure 4A the PixlMap probability image does not seem to be the exactly the same field or zoom as the original image. This should be explained or corrected.

Reviewer #3: Pennie et al. generated a computational tool to classify biomarkers in the multiplexed immunofluorescence images. They note that this tool is inspired by the accuracy of human intuition in identifying positive/negative features. This tool performs classification of biomarkers without the need to define cell boundaries by expanding the nucleus – which is also the primary novelty of this work. The efficacy of this tool was validated in human lung adenocarcinoma samples and murine liver samples.

The manuscript contains sufficient background information for an uninformed reader. Methods are described in sufficient detail and adequate data is shown to draw conclusions. I recommend acceptance of this manuscript. I have the following questions and recommendations.

1. Is fibroblast detection using PixlMap reliable? The authors seem to suggest that it needs further training to be able to detect fibroblasts. I suggest moving Figure 8 to the Supporting Information section if the authors think it needs more work.

2. In Figure 5B, it appears that PixlMap fails to identify some CD68+ cells. What is the accuracy and precision in this case?

3. In Figure 6, about 32% of true unclassifiable features are predicted as CD4+ by PixlMap? I think it will be important to identify biomarker types that have higher accuracy with PixlMap. For example, authors already mention that spindle shaped cells are not yet accurately identified by PixlMap. Similarly, are surface markers more accurately identified as opposed to cytoplasmic ones?

4. Provide a comparison of the weighted metrics (F1 score, accuracy, precision, and recall) of existing models with your model.

5. Lines 50 and 51: The authors only talk about tumor microenvironment as a driver to quantify multiple biomarkers simultaneously. There is a need to perform similar analyses in many other non-cancer studies. I suggest changing the sentence to incorporate other applications.

6. In the murine liver tissue, the accuracy of the model is significantly lower than in human tissue. The F1 scores are also about 0.3 at all probability thresholds. It is understandable that the model is not trained on murine tissues. However, please justify the inclusion of this data with weak metrics in the main manuscript.

7. In Figure 6H, the caption mentions “Confusion matrix comparing PixlMap and threshold-based phenotyping using mean pixel intensities of each marker”. However, data is shown for Manual annotation vs. threshold-based phenotyping. Please clarify.

**Do you want your identity to be public for this peer review?** For information about this choice, including consent withdrawal, please see our Privacy Policy

Reviewer #1: No

Reviewer #2: No

Reviewer #3: No

---

## [Author Response · Author response to Decision Letter 1]

26 Jun 2025

The Authors would like to take this opportunity to thank the Editor and Reviewers for the care and time taken in reviewing our work. We have responded to the feedback and suggestions received in detail in the attached document titled "Response to Reviewers".

---

## [Decision Letter · Decision Letter 1]

17 Aug 2025

PixlMap: A generalisable pixel classifier for cellular phenotyping in multiplex immunofluorescence images

PONE-D-25-00785R1

Dear Dr. Le Quesne,

We’re pleased to inform you that your manuscript has been judged scientifically suitable for publication and will be formally accepted for publication once it meets all outstanding technical requirements.

Kind regards,

Jordan Robin Yaron, Ph.D.

Academic Editor

PLOS ONE

Additional Editor Comments (optional):

Reviewers' comments:

Reviewer's Responses to Questions

**Comments to the Author**

Reviewer #3: All comments have been addressed

2. Is the manuscript technically sound, and do the data support the conclusions?

Reviewer #3: Yes

3. Has the statistical analysis been performed appropriately and rigorously?

Reviewer #3: Yes

4. Have the authors made all data underlying the findings in their manuscript fully available?

Reviewer #3: Yes

5. Is the manuscript presented in an intelligible fashion and written in standard English?

Reviewer #3: Yes

Reviewer #3: The authors have addressed all of my comments. I am satisfied with their response. I recommend acceptance of this manuscript in its revised version.

**Do you want your identity to be public for this peer review?** For information about this choice, including consent withdrawal, please see our Privacy Policy

Reviewer #3: No

---

## [Editor Report · Acceptance letter]

PONE-D-25-00785R1

PLOS ONE

Dear Dr. Le Quesne,

I'm pleased to inform you that your manuscript has been deemed suitable for publication in PLOS ONE. Congratulations! Your manuscript is now being handed over to our production team.

Kind regards,

on behalf of

Dr. Jordan Robin Yaron

Academic Editor

PLOS ONE